# Nanoscale Composite Protective Preparation for Cars Paint and Varnish Coatings

Andrey Vladimirovich Blinov [1], Andrey Ashotovich Nagdalian [1,2,*], Lyudmila Pavlovna Arefeva [3], Valery Nikolaevich Varavka [3], Oleg Vyacheslavovich Kudryakov [3], Alexey Alekseevich Gvozdenko [1], Alexey Borisovich Golik [1], Anastasiya Aleksandrovna Blinova [1], David Guramievich Maglakelidze [1], Dionis Demokritovich Filippov [1], Vyacheslav Anatolievich Lapin [1,4], Ekaterina Dmitrievna Nazaretova [5] and Mohammad Ali Shariati [6]

[1] Department of Physics and Technology of Nanostructures and Materials, North Caucasus Federal University, 355017 Stavropol, Russia
[2] Saint Petersburg State Agrarian University, 196601 Saint Petersburg, Russia
[3] Don State Technical University, 344000 Rostov-on-Don, Russia
[4] Federal Research Centre the Southern Scientific Centre of the Russian Academy of Sciences, 344006 Rostov-on-Don, Russia
[5] Specialized Educational Research Center, North Caucasus Federal University, 355017 Stavropol, Russia
[6] Department of Scientific Research, K.G. Razumovsky Moscow State University of Technologies and Management (The First Cossack University), 109004 Moscow, Russia
* Correspondence: geniando@yandex.ru

**Abstract:** In this work we have developed a nanoscale composite protective preparation for car paint and varnish coatings. We developed methods to obtain $SiO_2$-$TiO_2$, $TiO_2$-$ZrO_2$, $SiO_2$-$ZrO_2$ and $SiO_2$-$TiO_2$-$ZrO_2$ nanocomposites, which are crystallization centers for the formation of a nanoscale composite protective coating with certain morphology and roughness. The phase composition of the samples and stability in alkaline media were studied. It is shown that $SiO_2$-$TiO_2$-$ZrO_2$ nanocomposites with a content of titanium dioxide from 8%–9.5% and zirconium dioxide from 0.5%–2% exhibit complete insolubility in a highly alkaline medium, allow to form uniform structure on paint and varnish coatings, and protect the car surface from exposure to ultraviolet radiation. We determined the optimal composition of the preparation components for the formation of a nanoscale composite protective coating with hydrophobic properties and a wetting contact angle of more than 120 degrees: tetraethoxysilane $\leq$ 10 vol.%., titanium tetraisopropylate $\geq$ 2 vol.% and plant resin $\geq$ 8 vol.% Practical approval of the developed preparation on BMW X6 showed a pronounced hydrophobic effect. Evaluation of the stability of the nanoscale composite protective coating to the washing process showed that the developed preparation is able to maintain its hydrophobic properties for more than 150 washing cycles.

**Keywords:** protective coating; nanocomposites; paint and varnish coating; hydrophobicity; cars

## 1. Introduction

Human activity leads to ecological changes in the environment. The harmful impact of industry on the atmosphere leads to reactions including acid rain, sandstorms with metal particles, global warming etc. These factors have a destructive effect on car paint and varnish coatings [1–3]. Microparticles of metal and sand penetrate the paintwork structure and gradually destroy it. Acid rains corrode the paint varnish coatings, and the increased temperature leads to defects and reduced durability of the paintwork.

In addition to anthropogenic factors, bird and insect droppings, wood tar, residual tar, dirt, ice and ultraviolet radiation also have a negative impact on the cars paint and varnish coatings [4–6]. In this regard, the development of methods for the protection of cars paint and varnish coatings is a relevant task.

There are many different methods of protecting cars paint and varnish coatings. One of the most popular is to cover the car paintworks with a vinyl or polyurethane films protective [7]. Vinyl films are less expensive than polyurethane films, but they are harder and less susceptible to deformation, so they do not conform as well to body parts, and over time they can peel, crack, and burst. Polyurethane films are much more expensive than vinyl, but twice as thick and plastic. Because of their structural features, polyurethane films can tighten small scratches and scuffs when the body is heated [8,9]. Another popular method is the application of "liquid glass". This method is distinguished by the fact that this coating is well resistant to pollution and ultraviolet rays, because of its hydrophobic and antistatic properties. Unfortunately, "liquid glass" is not designed to protect against mechanical damage [10,11]. It is also worth noting the methods of protecting cars paint and varnish coatings protection such as treatment with various liquid waxes and polishes, covering of the cowl with armoring material, the installation of deflectors etc. [12–14].

However, one of the most relevant methods to protect the car paintwork is the ceramic coating, which in addition to its protective properties, also has aesthetic functions, increasing the color depth through a thick transparent layer [15]. One of the main disadvantages of modern ceramic coatings is their partial dissolution when washed with alkaline detergents and the presence of toxic phosphoric compounds.

Various means are used to protect a car's paintwork, including organic and silicone coatings, polystyrene complexes with various metals, and ceramic compounds. Protective coatings based on ceramic materials are the most promising, due to the simplicity of production, efficiency and durability, it should also be noted that the production of ceramic coatings is more cost-effective than other methods. All these advantages are achieved by using the sol–gel method for obtaining oxide materials, which are the basis for ceramic protective coatings [16–20]. Therefore, the purpose of this work is to develop and test a nanoscale composite protective preparation for car paint and varnish coatings using non-phosphoric components resistant to alkaline detergents.

## 2. Materials and Methods

### 2.1. Materials

Isopropyl alcohol ("HIMPROM" PJSC, Novocheboksarsk, Russia), Ethanol (JSC "BIOCHEM", Rasskazovo, Russia), n-butanol (JSC "BIOCHEM", Rasskazovo, Russia), Polymethylsilaxane liquid (JSC "GNIIKHTEOS", Moscow, Russia), Isoamyl alcohol (PKF "Iceberg AC", Yekaterinburg, Russia), Benzyl alcohol (LLC "AlbaKhim", Orenburg, Russia), Amyl alcohol (LLC "AlbaKhim", Orenburg, Russia), Propyl alcohol (LLC "AlbaKhim", Orenburg, Russia), tetraethoxysilane (JSC "GNIIKHTEOS", Moscow, Russia), titanium tetraisopropoxide (JSC "GNIIKHTEOS", Moscow, Russia), zirconium butylate (LLC "Setilim", Ufa, Russia), ammonia (LLC "METAKHIM", Moscow, Russia), potassium hydroxide (JSC Lenreactive, Saint Petersburg, Russia), plant resin (JSC Lenreactive, Saint Petersburg, Russia).

### 2.2. Methods of Component Synthesis

#### 2.2.1. Method of $SiO_2$-$TiO_2$ Synthesis

To synthesize the $SiO_2$-$TiO_2$ nanocomposite, we used a sol–gel method with tetraethoxysilane (TEOS) as the $SiO_2$ precursor, titanium tetraisopropylate as the $TiO_2$ precursor and an aqueous ammonia solution as the precipitator [21]. The synthesis was carried out in an alcoholic medium using ethanol. The method of obtaining $SiO_2$-$TiO_2$ nanocomposite included the next stages:

(1)  dissolution of tetraethoxysilane and titanium tetraisopropylate in ethanol in a volume ratio of 1:10;
(2)  addition of 12.5% aqueous ammonia solution to the reaction mass;
(3)  mixing of $SiO_2$-$TiO_2$ sol at 500 rpm for 24 h;
(4)  concentration and washing of the obtained samples by centrifugation at 3000 rpm for 10 min, (the process was repeated 5 times);

(5)    calcination of samples at 500 °C.

According to this method, a series of samples of $SiO_2$-$TiO_2$ nanocomposite containing 10% to 90% titanium dioxide were obtained for further experiments.

### 2.2.2. Method of $SiO_2$-$ZrO_2$ Synthesis

$SiO_2$-$ZrO_2$ nanocomposite was obtained by a similar method using tetraethoxysilane (TEOS) as a precursor of $SiO_2$, zirconium butylate as a precursor of $ZrO_2$ and an aqueous solution of ammonia as a precipitator. The method of obtaining $SiO_2$-$ZrO_2$ nanocomposite included the next stages:

(1)    dissolution of tetraethoxysilane in ethanol in a volume ratio 1:10, and zirconium butylate in distilled water;
(2)    mixing of the obtained solutions with the addition of 12.5% ammonia solution;
(3)    mixing of $SiO_2$-$ZrO_2$ sol at 500 rpm for 24 h;
(4)    concentration and washing of the obtained samples by centrifugation at 3000 rpm for 10 min, (the process was repeated 5 times);
(5)    calcination of samples at 500 °C.

According to this method, a series of samples of $SiO_2$-$ZrO_2$ nanocomposite containing 0.1% to 3% zirconium dioxide were obtained for further experiments.

### 2.2.3. Method of $TiO_2$-$ZrO_2$ Synthesis

$TiO_2$-$ZrO_2$ nanocomposite was obtained by a similar method using titanium tetraisopropylate as a precursor of $TiO_2$, zirconium butylate as a precursor of $ZrO_2$ and an aqueous solution of ammonia as a precipitator. The method of obtaining a $TiO_2$-$ZrO_2$ nanocomposite included the next stages:

(1)    dissolution of titanium tetraisopropylate in ethanol in a volume ratio of 1:10, and zirconium butylate in distilled water;
(2)    mixing of the obtained solutions with the addition of 12.5% ammonia solution;
(3)    mixing of $TiO_2$-$ZrO_2$ sol at 500 rpm for 24 h;
(4)    concentration and washing of the obtained samples by centrifugation at 3000 rpm for 10 min, (the process was repeated 5 times);
(5)    calcination of samples at 500 °C.

According to this method, a series of $TiO_2$-$ZrO_2$ nanocomposite samples containing 0.1% to 3% zirconium dioxide were obtained for further experiments.

### 2.2.4. Method of $SiO_2$-$TiO_2$-$ZrO_2$ Synthesis

$SiO_2$-$TiO_2$-$ZrO_2$ nanocomposite was obtained by the sol–gel method. Tetraethoxysilane, titanium tetraisopropylate and zirconium butylate were used as precursors. An aqueous ammonia solution was used as a precipitator. The method of obtaining $SiO_2$-$TiO_2$-$ZrO_2$ nanocomposite included the following stages:

(1)    dissolution of tetraethoxysilane and titanium tetraisopropylate in ethanol in a volume ratio of 1:10, and zirconium butylate in distilled water;
(2)    mixing of the obtained solutions with the addition of 12.5% ammonia solution;
(3)    mixing of $SiO_2$-$TiO_2$-$ZrO_2$ sol at 500 rpm for 24 h;
(4)    concentration and washing of the obtained samples by centrifugation at 3000 rpm for 10 min, (the process was repeated 5 times);
(5)    calcination of samples at 500 °C.

According to this method, a series of $SiO_2$-$TiO_2$-$ZrO_2$ nanocomposite samples containing 0.1% to 3% zirconium dioxide and 7% to 10% titanium dioxide were obtained for further experiments.

### 2.2.5. Method of Synthesis of Nanoscale Composite Preparation for Car Paint and Varnish Coatings

Our method of synthesis of nanoscale composite preparation for car paint and varnish coatings is shown schematically in Figure 1.

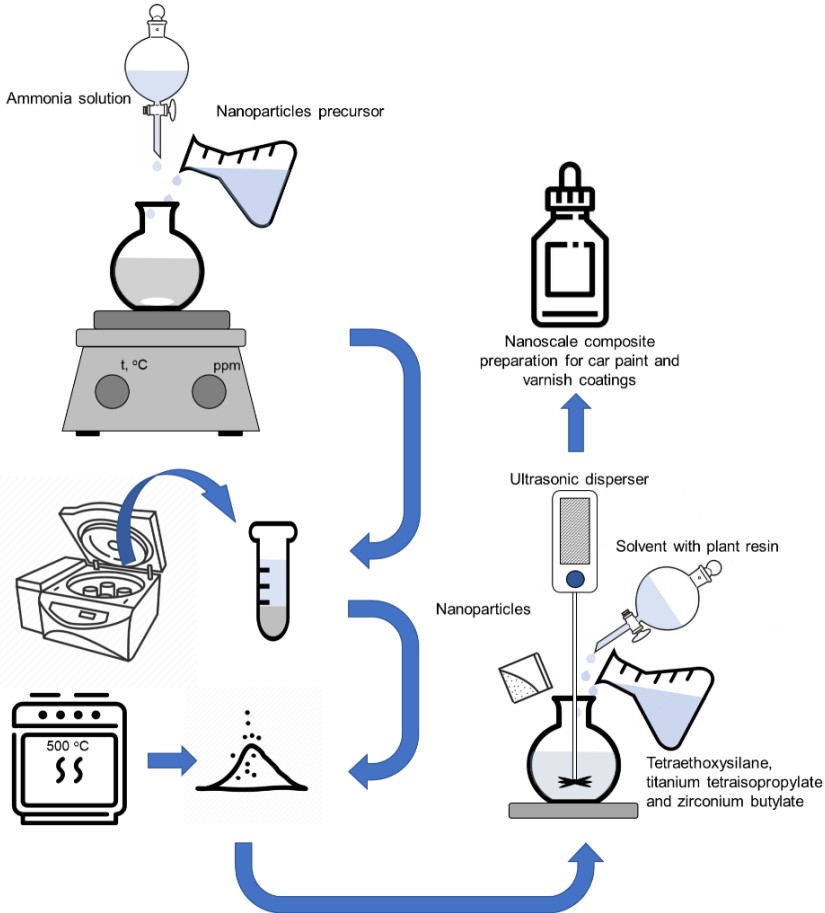

**Figure 1.** Scheme of nanoscale composite preparation synthesis.

Initially, sols containing $SiO_2$-$TiO_2$, $TiO_2$-$ZrO_2$, $SiO_2$-$ZrO_2$ and $SiO_2$-$TiO_2$-$ZrO_2$ nanocomposites seeding particles were formed as crystallization centers for the formation of a nanoscale composite protective coating with a certain morphology and roughness. The synthesis of nanocomposites was carried out according to Sections 2.2.1–2.2.4. The washed samples of nanocomposites were dispersed in a solvent with plant resin and mixed for 15 min. Dispersion was carried out using Ultrasonic homogenizer UP400S (Hielscher Ultrasonics GmbH, Teltow, Germany) with set parameters: Sono-rod H3 (titanium, 100 mm), processing power 200 W, frequency 24 kHz, amplitude 60%, pulsation 70%, and treatment duration 30–60 s. The achieved level of particle dispersion was 50–150 nm. The physicochemical properties of the plant resin are presented in Table 1 [22]. In the final step, tetraethoxysilane, titanium tetraisopropylate and zirconium butylate were added to the reaction system and also mixed for 15 min.

**Table 1.** The physicochemical properties of plant resin.

| Name of the Indicator | Meaning |
| --- | --- |
| Appearance and smell | Transparent volatile liquid with a characteristic odor without sediment and water |
| Density at 20 °C, g/cm$^3$ | 0.855–0.863 |
| Refractive index | 1.465–1.472 |

**Table 1.** *Cont.*

| Name of the Indicator | Meaning |
|---|---|
| Color intensity | No more intense than the coloring of the equal-height volume of the comparison solution N 1 |
| The top of the tank is in the maximum temperature at 101325 Pa (760 mm PT.St.), %:<br>up to 155 °C<br>up to 170 °C | <br><br>Absent<br>≥92 |
| Mass fraction of sum α- and β-pinen, % | ≥60 |
| Acid number, mg KOH per 1 g of product | ≤0.5 |
| Mass fraction of non-volatile residue, % | ≤0.5 |

2.2.6. Practical Approval of the Nanoscale Composite Preparation for the Car Paint and Varnish Coatings

The practical approval of the nanoscale composite preparation for the car paint and varnish coatings was carried out on the basis of LLC "Research and Production Company PRIDE" (Stavropol, Russia). First, a sol containing seed particles of $SiO_2$-$TiO_2$-$ZrO_2$ nanocomposite was formed. The synthesis of nanocomposites was carried out according to Section 2.2.4. The washed samples of nanocomposites were dispersed in isopropanol and mixed for 15 min with the addition by 8 vol.% of plant resin. In the final stage, tetraethoxysilane, titanium tetraisopropylate and zirconium butylate were added to the reaction system and mixed for 15 min. The content of tetraethoxysilane was 10 vol.%, titanium tetraisopropylate −2 vol.%

The obtained samples from the nanoscale composite preparation for car paint and varnish coatings were transparent, homogeneous, slightly oily liquids with low specific odor and not pungent. The samples were packaged in 50 mL brown glass vials with a chemically inert black plastic screw cap.

*2.3. General Methods and Equipment*

– Scanning electron microscopy using a MIRA-LMH scanning electron microscope with the AZtecEnergy Standard/X-max 20 (standard) elemental composition determination system from Tescan. The parameters of the measurement were as follows:

- Voltage 10 kV.
- Work Distance 4.9 mm.
- In-Beam SE detector.

– Multi-angle particle size analyzer Photocor Complex (LLC "Photocor", Moscow, Russia). Processing of the results was carried out using the DynaLS software.

– X-ray diffraction analysis on an Empyrean diffractometer (PANalytical, Almeo, The Netherlands). The following measuring parameters were used:

- Copper cathode (wavelength 1.54 Å).
- Measurement range 10–90 2θ°.
- Sampling frequency: 0.01 2θ°.

– SF-56 spectrophotometer with a prefix for measuring diffuse reflection spectra (OKB "Spectrum", St. Petersburg, Russia).

– LAUDA LSA100 surface analyzer (LAUDA DR. R. WOBSER GMBH & CO. KG, Lauda-Konigshofen, Germany).

– Climate chamber of heat-cold-moisture VIKAM-1000/1 ("Techno-Priest", Moscow, Russia).

### 2.4. Investigation of the Stability of Samples of Nanoscale Composites and Their Components in an Alkaline Medium

This experiment took place in two stages:

(1) a preliminary experiment of studying the stability of silicon dioxide sample, which is a component of the $SiO_2$-$TiO_2$, $SiO_2$-$ZrO_2$ and $SiO_2$-$TiO_2$-$ZrO_2$ nanocomposites and makes up the most of their phase composition;

(2) study of the stability of $SiO_2$-$TiO_2$, $TiO_2$-$ZrO_2$, $SiO_2$-$ZrO_2$ and $SiO_2$-$TiO_2$-$ZrO_2$ nanocomposites.

The stability of the nanocomposites and their components in alkaline medium was studied according to the following procedure:

(1) A total of 0.2 g of the nanocomposites was placed in 20 mL of potassium hydroxide solution;

(2) the samples were kept for 180 min;

(3) Three mL of the solution was taken every 30 min and analyzed by dynamic light scattering on the Photocor Complex installation.

To study the stability of a silicon dioxide sample in an alkaline medium, we prepared 0.1, 1, 2.5 and 5 M solutions of potassium hydroxide. To prepare solutions, the required amount of potassium hydroxide was weighted on analytical scales and transferred to a heat-resistant measuring cup. Then 100 mL of distilled water was added to prevent the solution from heating above 70 °C.

Five M solution of potassium hydroxide in alkaline medium was prepared to investigate the stability of the samples of $SiO_2$-$TiO_2$, $TiO_2$-$ZrO_2$, $SiO_2$-$ZrO_2$ and $SiO_2$-$TiO_2$-$ZrO_2$ nanocomposites. Samples of the following compositions were studied: $SiO_2$-$TiO_2$ with titanium dioxide content from 10%–90%, $TiO_2$-$ZrO_2$ with zirconium dioxide content from 0.1%–3%, $SiO_2$-$ZrO_2$ with zirconium dioxide content from 0.1%–3%, $SiO_2$-$TiO_2$-$ZrO_2$ with a titanium dioxide content from 7%–10% and dioxide zirconium from 0.1%–3%.

### 2.5. Optimization of the Method of Synthesis of the Nanoscale Composite Preparation for the Car Paint and Varnish Coatings

A multifactorial experiment was carried out to optimize the synthesis method of the nanoscale composite preparation for the car paint and varnish coatings. The following parameters were considered as input parameters: volume concentration of plant resin ($\eta_1$ (plant resin)), volume concentration of tetraethoxysilane ($\eta_2$ (tetraethoxysilane)) and the volume concentration of titanium tetraisopropylate ($\eta_3$ (titanium tetraisopropylate)). The levels of variation of the main components are presented in Table 2.

**Table 2.** The levels of variation of the main components.

| Parameter | Levels of Variable Variation | | |
|---|---|---|---|
| $\eta_1$ (plant resin), vol.% | 0.5 | 2.5 | 4.5 |
| $\eta_2$ (tetraethoxysilane), vol.% | 5 | 15 | 25 |
| $\eta_3$ (titanium tetraisopropylate), vol.% | 0.5 | 2 | 3.5 |

Xanthones: a class of heterocyclic compounds with anticancer potential Table 2. Levels of variation of the main variable parameters.

It is important to note that the content of zirconium butylate in the samples remained constant and amounted to 0.1% weight mass. The wetting contact angle, which characterizes the hydrophobicity of the surface and the degree of interaction of polluting liquids, suspensions and emulsions with the surface of the paint and varnish coating, was considered as the initial parameter [23]. When the wetting contact angle $\theta \leq 90°$, the surface is hydrophilic, which leads to rapid contamination of paint and varnish coatings because the contaminating liquid interacts better with the car surface. The hydrophobic surface is characterized by wetting contact angle $\theta > 90°$ [23]. When in contact with a hydrophobic

surface, the contaminating liquid is quickly removed from the surface, keeping the car's paint and varnish "clean" longer [24]. These types of surfaces are illustrated in Figure 2.

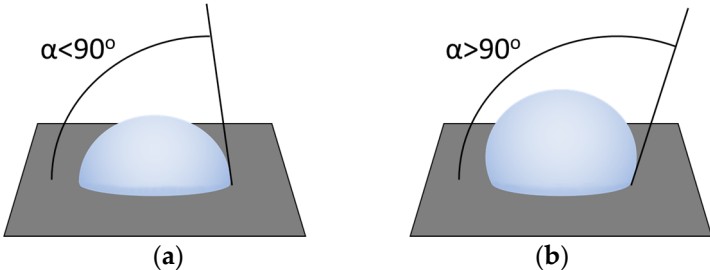

**Figure 2.** Hydrophilic (**a**) and hydrophobic (**b**) surfaces.

The experiment planning matrix is presented in Table 3.

**Table 3.** Experiment planning matrix with numerical values of variable parameters for each experiment.

| Experiment 1 | - | Experiment 2 | - | Experiment 3 | - |
|---|---|---|---|---|---|
| $\eta_1$, vol.% | 0.5 | $\eta_1$, vol.% | 0.5 | $\eta_1$, vol.% | 0.5 |
| $\eta_2$, vol.% | 5 | $\eta_2$, vol.% | 15 | $\eta_2$, vol.% | 25 |
| $\eta_3$, vol.% | 0.5 | $\eta_3$, vol.% | 2 | $\eta_3$, vol.% | 3.5 |
| **Experiment 4** | **-** | **Experiment 5** | **-** | **Experiment 6** | **-** |
| $\eta_1$, vol.% | 2.5 | $\eta_1$, vol.% | 2.5 | $\eta_1$, vol.% | 2.5 |
| $\eta_2$, vol.% | 5 | $\eta_2$, vol.% | 15 | $\eta_2$, vol.% | 25 |
| $\eta_3$, vol.% | 2 | $\eta_3$, vol.% | 3.5 | $\eta_3$, vol.% | 0.5 |
| **Experiment 7** | **-** | **Experiment 8** | **-** | **Experiment 9** | **-** |
| $\eta_1$, vol.% | 4.5 | $\eta_1$, vol.% | 4.5 | $\eta_1$, vol.% | 4.5 |
| $\eta_2$, vol.% | 5 | $\eta_2$, vol.% | 25 | $\eta_2$, vol.% | 25 |
| $\eta_3$, vol.% | 3.5 | $\eta_3$, vol.% | 0.5 | $\eta_3$, vol.% | 2 |

Characteristics of the solvents used for the synthesis of the drug are presented in Table 4 [25,26].

**Table 4.** Characteristics of solvents used for the preparation synthesis.

| Solvent | Formula | Price $/L | MAC *, mg/m$^3$ | Odor Acridity | Hazard Class |
|---|---|---|---|---|---|
| isopropanol | $C_3H_8O$ | 3.73 | 10 | ++ | 3 |
| ethanol | $C_2H_5OH$ | 0.94 | 10 | + | 3 |
| n-butanol | $C_4H_{10}O$ | 6.75 | 10 | + | 3 |
| polymethylsiloxane liquid | $(CH_3)_3SiO\,[SiO(CH_3)_2]_n\,Si(CH_3)_3$ | 4.05 | 10 | - | 5 |
| isoamyl alcohol | $C_5H_{12}O$ | 4.57 | 10 | +++ | 3 |
| benzyl alcohol | $C_7H_8O$ | 5.13 | 5 | +++ | 3 |
| amyl alcohol | $C_5H_{11}OH$ | 5.81 | 10 | +++ | 3 |
| propanol | $C_3H_7OH$ | 2.09 | 10 | + | 3 |

* MAC—maximum allowable concentration.

Statistical processing of the obtained data was carried out in the Statistica 12.0 software [27]. A package of application Neural Statistica Network software was used to construct the response surfaces [28]. The architecture of formed neural network is shown in Figure 3.

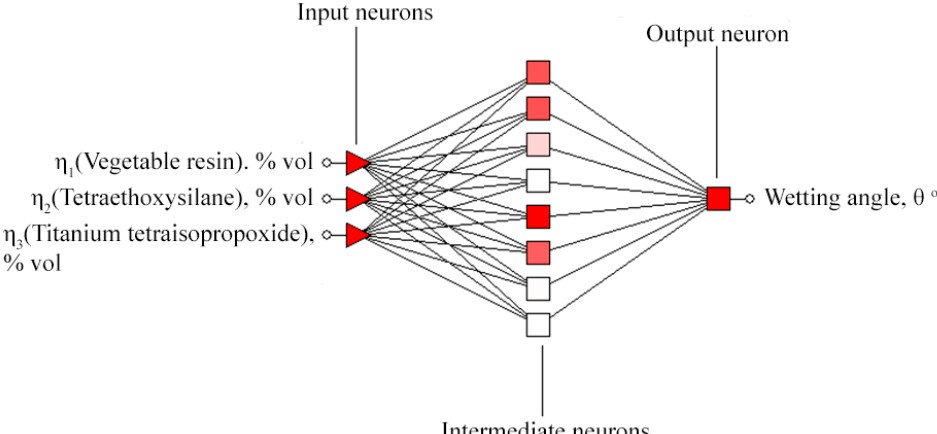

**Figure 3.** Architecture of a multilayer perceptron formed to study the effect of the component composition of the preparation on the formation of a nanoscale composite protective coating.

*2.6. Method of Formation of Nanoscale Composite Protective Preparation for Car Paint and Varnish Coatings*

The formation of a nanoscale composite protective preparation for car paint and varnish coatings was carried out according to the following method:

(1) the car paint was cleaned from metal inclusions and other types of contamination using the following technological operations:

    – two-phase surface washing;
    – removal of metal particles (application of a tool to removal of metal inclusions) [29];
    – car paintwork washing.

(2) polishing the car paintwork;
(3) degreasing the car paintwork;
(4) ten mL of the preparation, obtained according to Sections 2.2.5 and 2.2.6, were applied to the car paintwork;
(5) finally, complete drying of the nanoscale composite protective layer was waited.

*2.7. Method for Determining the Wetting Contact Angle*

A total of 0.1 mL of distilled water was applied to the sample surface with a nanoscale composite protective coating using a micropipette. The wetting contact angle was determined using the LAURA LA 100 surface analyzer (LAUDA DR. R. WOBSER GMBH & CO. KG, Lauda-Konigshofen, Germany). The study was carried out three times.

*2.8. Investigation of the Stability of a Nanocomposite Protective Coating*

To study the stability of the nanoscale composite protective preparation for car paint and varnish coatings, metal plates with a size of $10 \times 10$ mm coated with nanoscale composite protective preparation for car paint and varnish coatings were placed in a climate chamber with the ability to regulate temperature and humidity at various parameters for 24 h, after previously measured the wetting contact angle. After modeling the extreme weather conditions, the coated plates were removed, brought to room temperature, and the wetting contact angle was re-measured

*2.9. Analysis of the Stability of a Nanoscale Composite Protective Coating to the Washing Process*

The stability of a nanoscale composite protective coating toward the washing process was evaluated on the surface of a BMW X6 car using the following method:

(1) The car surface with a nanoscale composite protective coating was treated with car shampoo using a foam generator;
(2) waited 1–2 min;
(3) The car shampoo was washed off with osmotic water;

(4)    the wetting contact angle was determined according to Section 2.7.

This procedure was repeated 250 times.

## 3. Results and Discussion

### 3.1. Investigation of SiO$_2$-TiO$_2$, TiO$_2$-ZrO$_2$, SiO$_2$-ZrO$_2$ and SiO$_2$-TiO$_2$-ZrO$_2$ Nanocomposite Samples

Initially, the phase composition of nanocomposite samples was studied: SiO$_2$-TiO$_2$ with titanium dioxide content of 10% to 90%; TiO$_2$-ZrO$_2$ with a zirconium dioxide content of 0.1% to 3%; SiO$_2$-ZrO$_2$ with a zirconium dioxide content of 0.1% to 3%; SiO$_2$-TiO$_2$-ZrO$_2$ with titanium dioxide content of 7%–10% and zirconium dioxide from 0.1%–3%. The obtained diffractograms are shown in Figures 4–7 and the Supplementary Material.

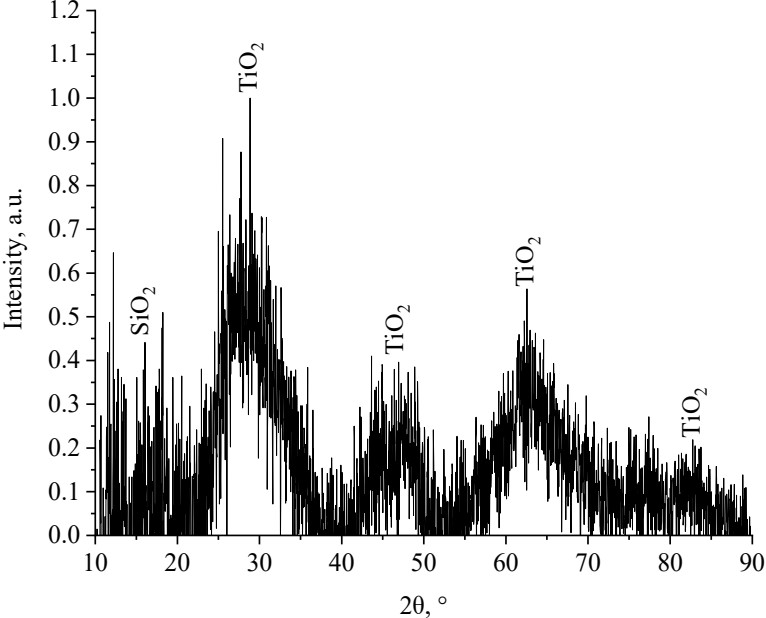

**Figure 4.** Diffractogram of a SiO$_2$-TiO$_2$ sample containing 10% SiO$_2$.

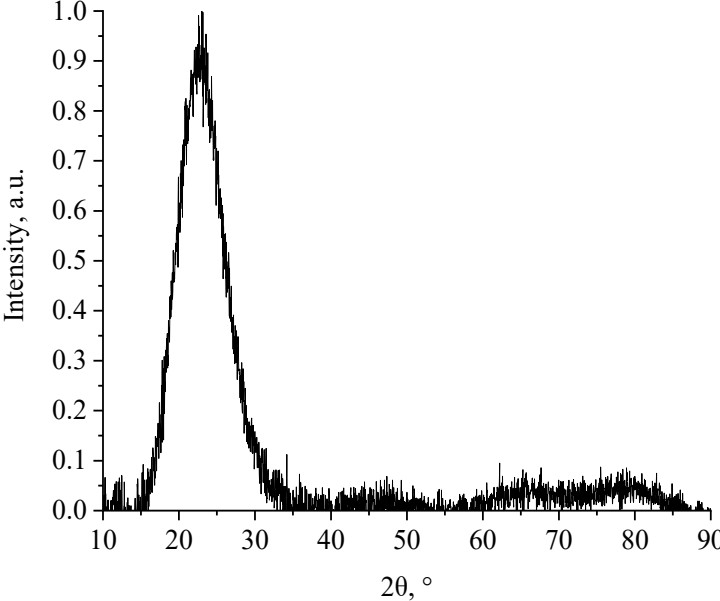

**Figure 5.** Diffractogram of a SiO$_2$-ZrO$_2$ sample containing 1% ZrO$_2$.

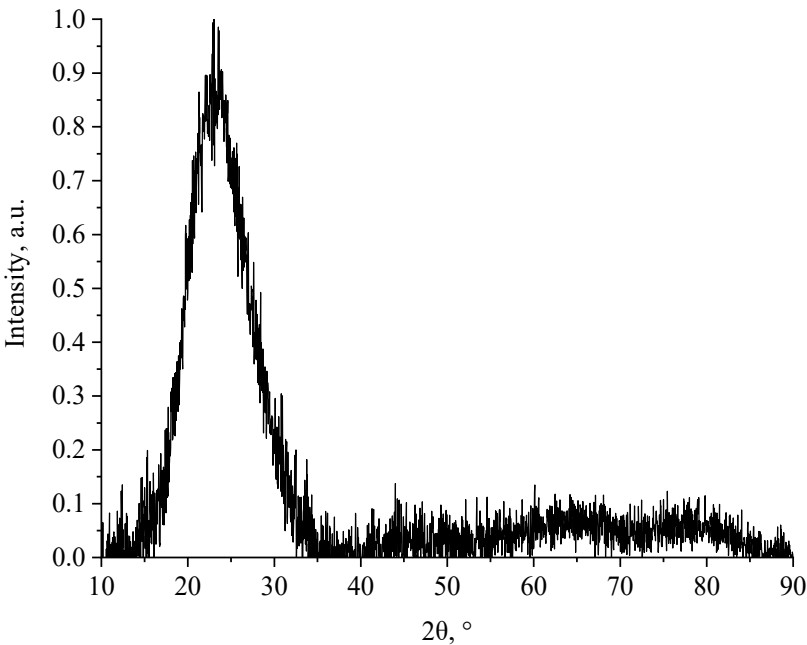

**Figure 6.** Diffractogram of a $SiO_2$-$TiO_2$-$ZrO_2$ sample containing 1% $ZrO_2$ and 9% $TiO_2$.

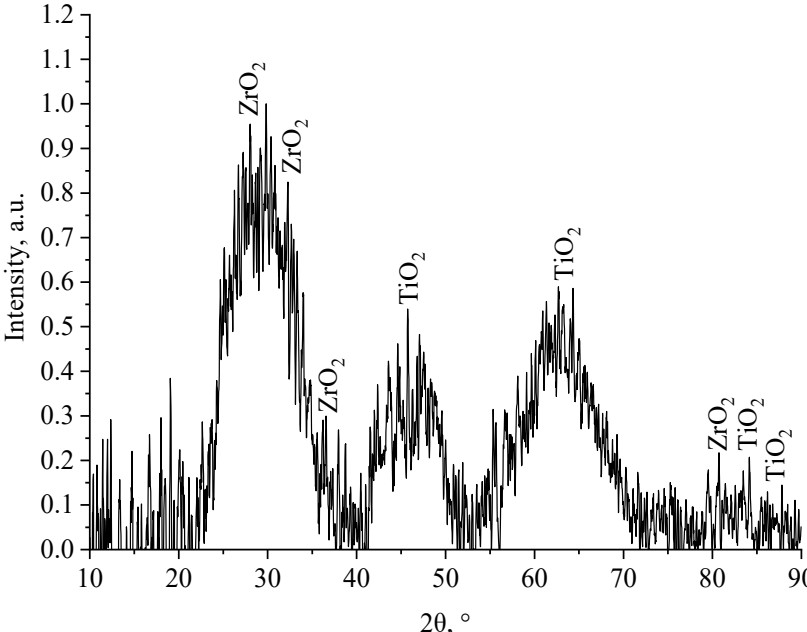

**Figure 7.** Diffractogram of a $TiO_2$-$ZrO_2$ sample containing 1% $ZrO_2$.

We found low-intensity bands in the diffractograms of $SiO_2$-$TiO_2$, $SiO_2$-$ZrO_2$ and $SiO_2$-$TiO_2$-$ZrO_2$ samples, indicating that synthesized nanocomposites have an amorphous structure.

It is important to note that with an increase in titanium dioxide in a series of $SiO_2$-$TiO_2$ samples, a transition from an amorphous to a crystalline structure occurs. The $TiO_2$-$ZrO_2$ nanocomposite samples are crystalline and contain non-stoichiometric zirconium oxide with a tetragonal crystal lattice [30,31], as well as tetragonal modification titanium dioxide in the anatase type structure [32–34].

Since the nanoscale composite protective preparation for car paint and varnish coatings will be periodically exposed to highly alkaline environments during car surface washing to remove various contaminants [35], the stability of nanocomposites samples and their components in an alkaline environment was further investigated.

Initially, we investigated the stability of silicon dioxide, which constitutes a major part of the phase composition of the prepared samples. The obtained kinetic curves of the $SiO_2$ scattering intensity as a function of time and active acidity of the medium are presented in Figure 8.

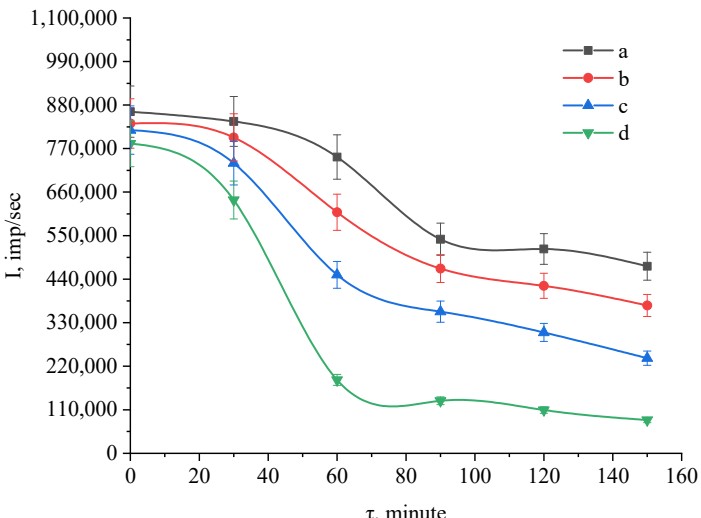

**Figure 8.** Kinetic curves of the scattering intensity of the $SiO_2$ sample in KOH solutions: a—0.1 M, b—1 M, c—2.5 M, d—5 M.

Analysis of the data obtained, showed that the scattering intensity of silicon dioxide samples decreases with the exposure time in alkaline medium and the most significant changes are observed in the 5 M KOH solution, when the dependence of the scattering intensity on time becomes exponential. This decrease in scattering intensity is due to the dissolution of silicon dioxide in accordance to the reaction equation [36]:

$$SiO_2 + KOH \rightarrow KHSiO_3$$

In this regard, the stability of all nanocomposites samples was carried out in a 5 M KOH solution. The obtained kinetic curves are presented in Figures 9–12.

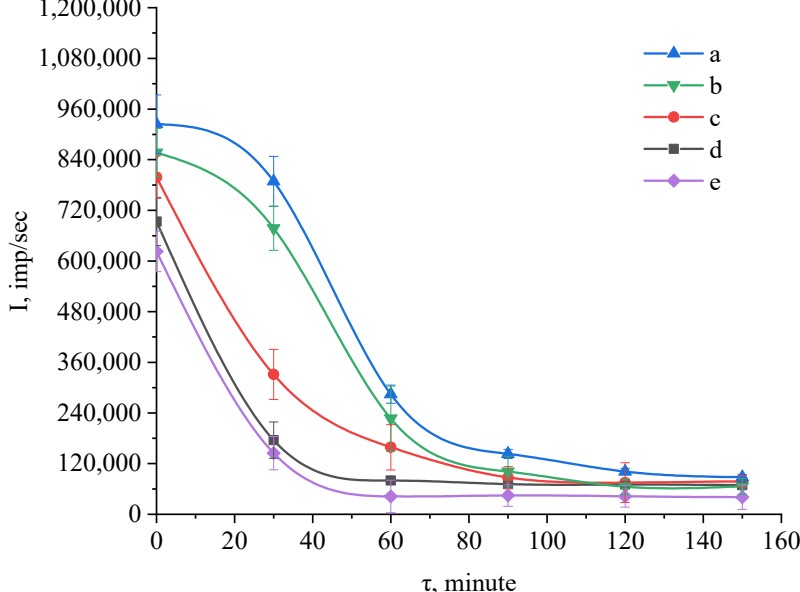

**Figure 9.** Kinetic curves of scattering intensity of $TiO_2$-$ZrO_2$ nanocomposite samples: a—0.1% $ZrO_2$, b—0.5% $ZrO_2$, c—1% $ZrO_2$, d—2% $ZrO_2$, e—3% $ZrO_2$.

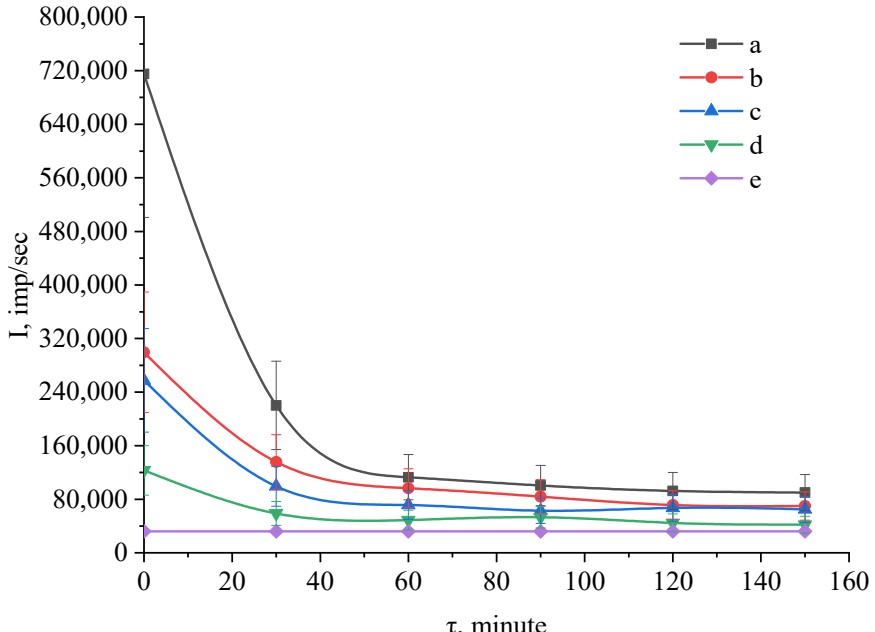

**Figure 10.** Kinetic curves of scattering intensity of SiO$_2$-ZrO$_2$ nanocomposite samples: a—0.1% ZrO$_2$, b—0.5% ZrO$_2$, c—1% ZrO$_2$, d—2% ZrO$_2$, e—3% ZrO$_2$.

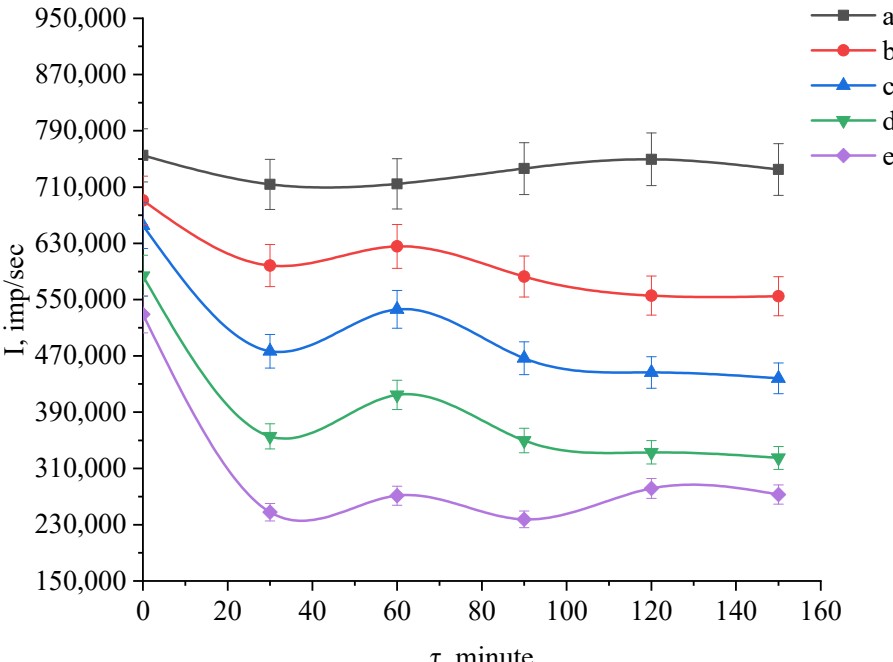

**Figure 11.** Kinetic curves of scattering intensity of SiO$_2$-TiO$_2$ nanocomposite samples: a—10% SiO$_2$ and 90% TiO$_2$; b—30% SiO$_2$ and 70% TiO$_2$; c—50% SiO$_2$ and 50% TiO$_2$; d—70% SiO$_2$ and 30% TiO$_2$; e—90% SiO$_2$ and 10% TiO$_2$.

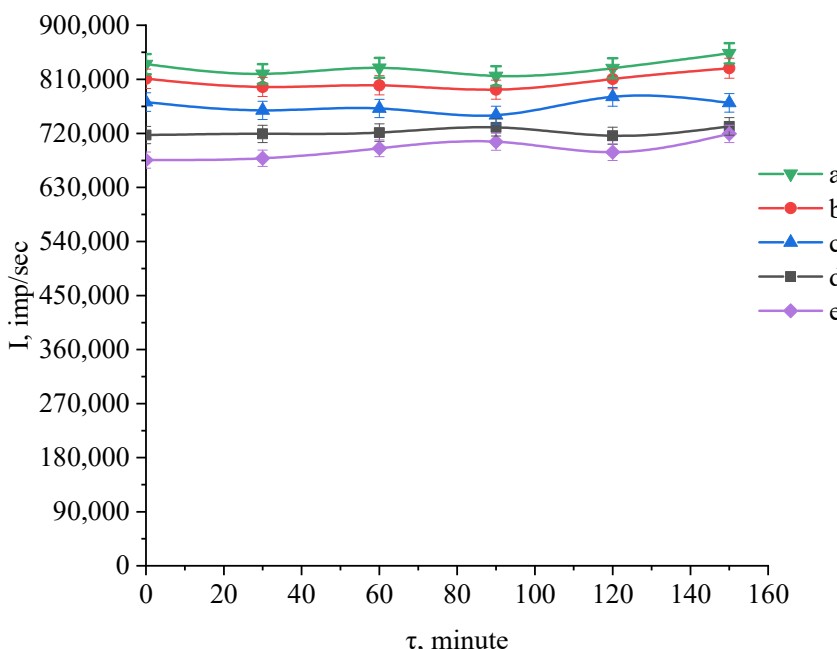

**Figure 12.** Kinetic curves of scattering intensity of $SiO_2$-$TiO_2$-$ZrO_2$ nanocomposite samples: a—0.1% $ZrO_2$ and 9.9% $TiO_2$; b—0.5% $ZrO_2$ and 9.5% $TiO_2$; c—1% $ZrO_2$ and 9% $TiO_2$; d—2% $ZrO_2$ and 8% $TiO_2$; e—3% $ZrO_2$ and 7% $TiO_2$.

We found that the scattering intensity of aqueous suspensions of $SiO_2$-$ZrO_2$ and $TiO_2$-$ZrO_2$ nanocomposites decreases significantly in a strongly alkaline medium at pH $\geq$ 13 [37–39]. It is established that the lower the concentration of zirconium dioxide in these composites, the faster they dissolve.

Kinetic curves of the scattering intensity of $SiO_2$-$TiO_2$ nanocomposite samples from the exposure time in 5 M KOH solution are shown in Figure 10.

Following the analysis of experimental data, it was found that the dispersion intensity of the aqueous suspension of $SiO_2$-$TiO_2$ nanocomposite sample with 90% titanium dioxide concentration of 90% does not change significantly from the exposure time of 5 M KOH, which indicates the absence of dissolution of this nanocomposite sample [40–42].

For a $TiO_2$ concentration of 10%–70%, a drop in the intensity of the suspension dispersion is observed from the exposure time of 5 M KOH, indicating dissolution of the nanocomposite samples. The largest intensity drops (270,000 pulses/s) was detected in a sample with 10% $TiO_2$. The nanocomposites were also studied by the energy dispersive X-ray spectroscopy (EDS) method. The EDS results were presented and discussed in our previous work [21]. Generally, our research showed that in addition to silicon, titanium, and zirconium dioxides, the nanocomposites included oxo-, hydroxo-, and aqua complexes of the corresponding elements. The kinetic curves of the scattering intensity of $SiO_2$-$TiO_2$-$ZrO_2$ nanocomposite samples as a function of exposure time in 5 M KOH solution are shown in Figure 11.

We found that the diffusion intensity of aqueous suspensions of $SiO_2$-$TiO_2$-$ZrO_2$ nanocomposite samples does not show statistically significant changes. This fact can be explained by the formation of solid solutions with unique physicochemical properties that can form these oxides at these concentrations.

### 3.2. Selection of Components for the Synthesis of the Protective Preparation for Car Paint and Varnish Coatings

Samples of $SiO_2$-$TiO_2$, $TiO_2$-$ZrO_2$, $SiO_2$-$ZrO_2$, $SiO_2$-$ZrO_2$ and $SiO_2$-$TiO_2$-$ZrO_2$ nanocomposites were used as seed particles in the preparation for the formation of a nanoscale composite protective coating. The experimental samples of the preparation were obtained according to Section 2.2.5.

For the experiment, 10 cm × 10 cm metal plates were prepared, on which a paint and varnish coating was applied using all the necessary technological operations, obtaining an experimental surface that completely imitates the paint of a car [43]. The preparation for the formation of a nanoscale composite protective coating was applied to the surface of the plates according to Section 2.6 and examined by diffuse light reflection and scanning electron microscopy (SEM). The obtained diffuse light reflection spectra are shown in Figure 13.

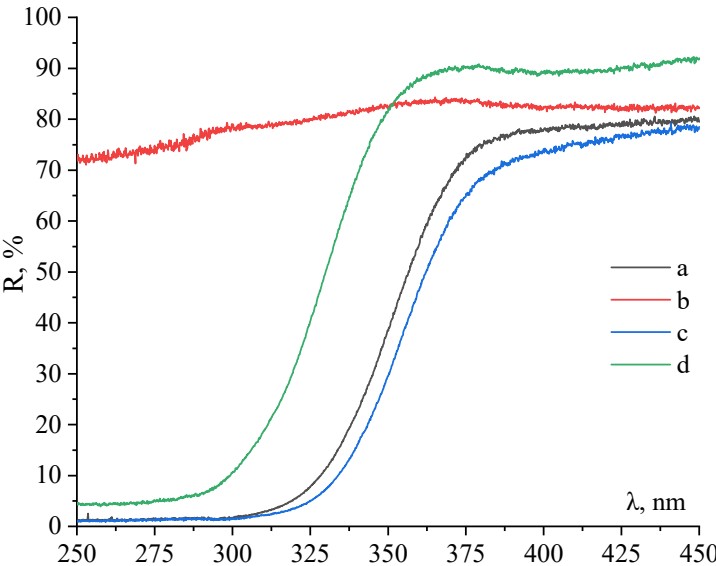

**Figure 13.** Diffuse reflection spectra of nanoscale composite protective coating samples obtained using nanocomposites: (a) $SiO_2$-$TiO_2$, (b) $SiO_2$-$ZrO_2$, (c) $TiO_2$-$ZrO_2$, (d) $SiO_2$-$TiO_2$-$ZrO_2$.

We have established that in the samples of a nanoscale composite protective coating, almost complete reflection (R > 73%) is observed both in the region of near- and medium UV-radiation from 250 to 400 nm, and in the visible spectrum region from 400 to 450 nm, which is due to the dielectric nature of prepared material [44].

Since the $SiO_2$-$TiO_2$, $TiO_2$-$ZrO_2$ and $SiO_2$-$TiO_2$-$ZrO_2$ nanocomposites samples contain titanium dioxide, which is a semiconductor material, the diffuse reflection spectra of these samples have the form characteristic of semiconductors [45]. In the medium ultraviolet region from 250 to 300 nm, the light absorption is observed by the samples of a nanoscale composite protective coating obtained using $SiO_2$-$TiO_2$, $TiO_2$-$ZrO_2$ and $SiO_2$-$TiO_2$-$ZrO_2$ nanocomposites (R ≤ 5%).

In the near ultraviolet region, the reflection coefficient increases from R = 2%–5% to R = 72%–90%. In the visible spectrum from 400–450 nm, the reflection was almost complete (R > 70%). Thus, we established that the developed nanoscale composite protective coating developed from $SiO_2$-$TiO_2$, $TiO_2$-$ZrO_2$, $SiO_2$-$ZrO_2$ and $SiO_2$-$TiO_2$-$ZrO_2$ nanocomposites absorbs or reflects near- and medium UV-radiation and visible light in the region of 400–450 nm, thereby protecting the car paint surface. The study of the microstructure of the composite protective coating samples was carried out by SEM. The obtained micrographs of the samples are shown in Figures 14–17.

SEM micrography analysis showed that the samples of nanoscale composite protective coating based on $SiO_2$-$TiO_2$, $TiO_2$-$ZrO_2$ and $SiO_2$-$ZrO_2$ nanocomposites have poor adhesion to the paint and varnish surface, as their structure is characterized by heterogeneity, irregularities in the form of folds and delamination, and inclusions of gas bubbles. A sample of a nanoscale composite protective coating based on $SiO_2$-$TiO_2$-$ZrO_2$ nanocomposite is distinguished by the uniformity of the structure, which is represented by microspheres with a diameter from 100 to 500 nm.

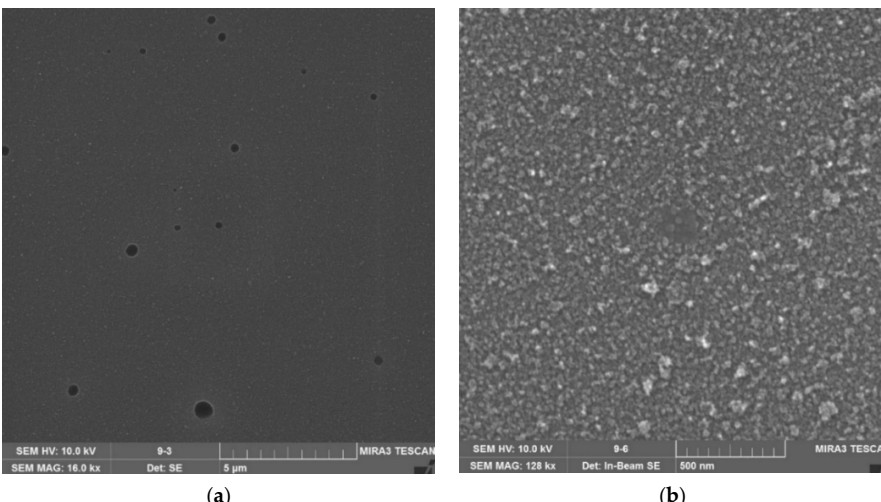

**Figure 14.** SEM micrography of a sample of a nanoscale composite protective coating obtained using TiO$_2$-ZrO$_2$ nanocomposite: (**a**) zoom ×16,000, (**b**) zoom ×128,000.

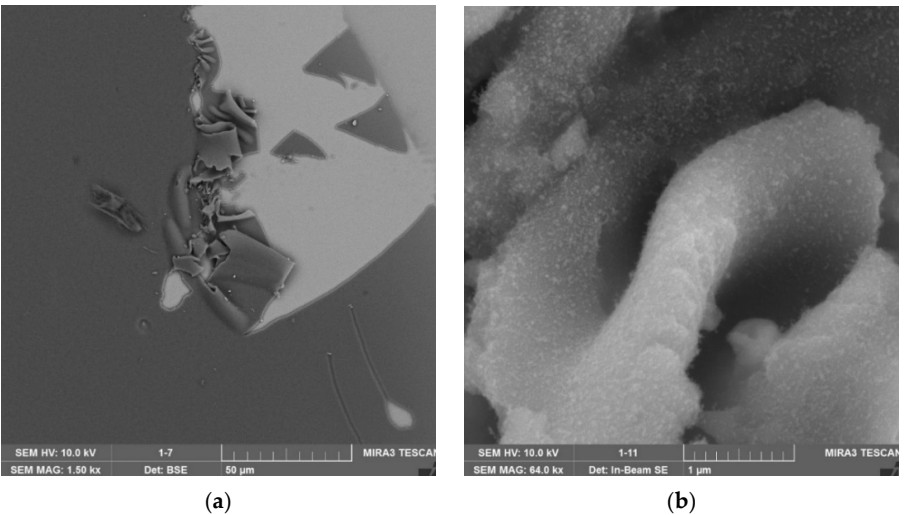

**Figure 15.** SEM micrography of a sample of a nanoscale composite protective coating obtained using SiO$_2$-ZrO$_2$ nanocomposite: (**a**) zoom ×1500, (**b**) zoom ×64,000.

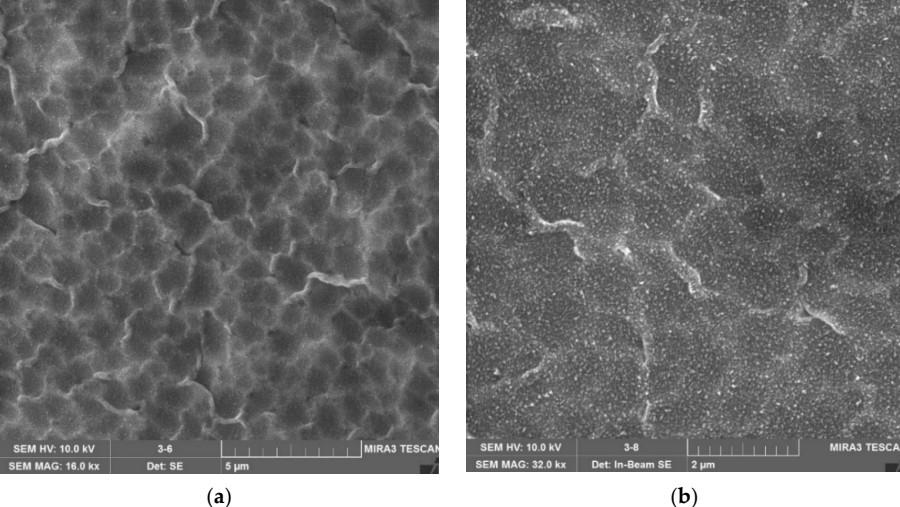

**Figure 16.** SEM micrography of a sample of a nanoscale composite protective coating obtained using SiO$_2$-TiO$_2$ nanocomposite: (**a**) zoom ×18,000, (**b**) zoom ×32,000.

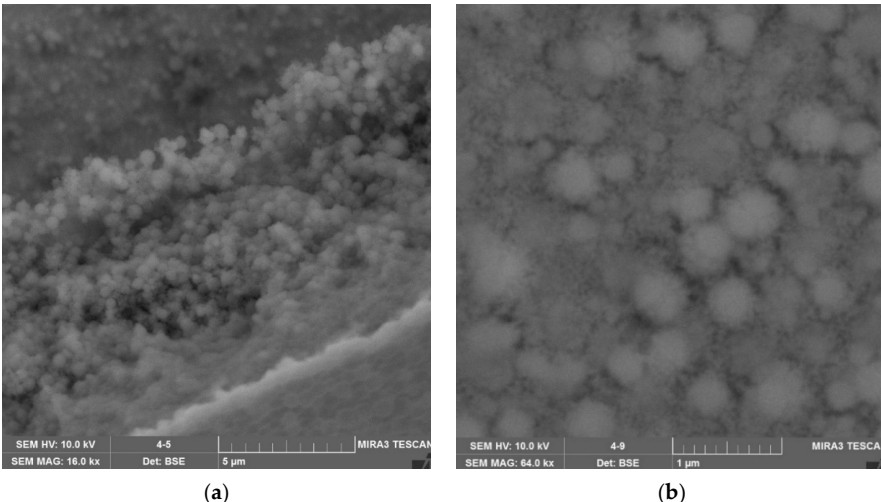

**Figure 17.** SEM micrography of a sample of a nanoscale composite protective coating obtained using SiO$_2$-TiO$_2$-ZrO$_2$ nanocomposite: (**a**) zoom ×16,000, (**b**) zoom ×64,000.

Based on the research conducted, we have determined that the optimal nanocomposite for obtaining a nanoscale composite protective coating is SiO$_2$-TiO$_2$-ZrO$_2$ with a content of titanium dioxide from 8%–9.5% and zirconium dioxide from 0.5%–2%, which does not dissolve in a highly alkaline medium, allows the formation of a uniform structure on the surface of the paint and varnish coating and protects the car surface from exposure to ultraviolet radiation.

In the next stage, a solvent was selected to obtain a preparation for the formation of a nanoscale composite protective coating. The synthesis of the preparation was carried out according to Section 2.2.5. SiO$_2$-TiO$_2$-ZrO$_2$ nanocomposite samples were used as seed particles. The following compounds were considered as a solvent: isopropanol, ethanol, n-butanol, polymethylsiloxane liquid, isoamyl alcohol, benzyl alcohol, amyl alcohol, propanol.

Photos of prepared samples of the preparation are shown in Figure 18.

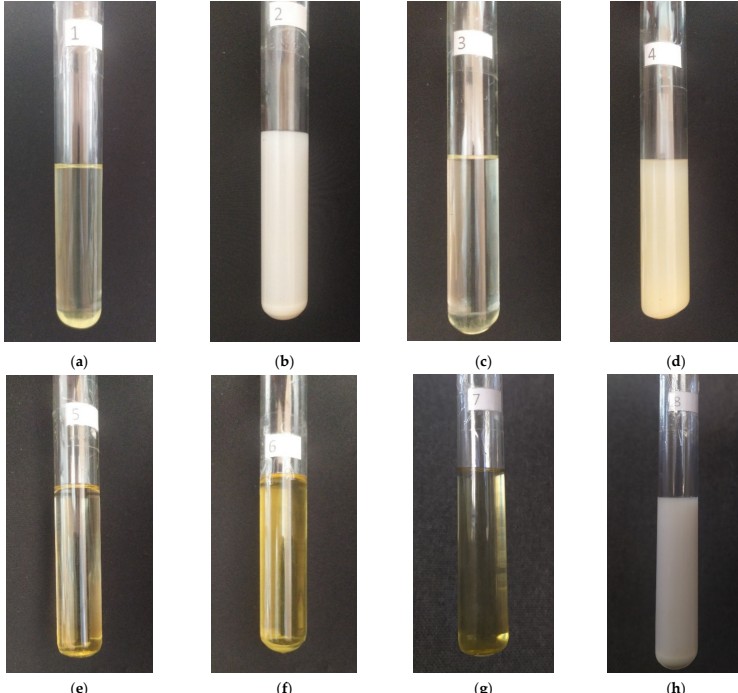

**Figure 18.** Samples photos of the preparation for the formation of a nanoscale composite protective coating obtained using various solvents: (**a**) isopropanol, (**b**) ethanol, (**c**) n-butanol, (**d**) polymethyl-siloxane liquid, (**e**) isoamyl alcohol, (**f**) benzyl alcohol, (**g**) amyl alcohol, (**h**) propanol.

As shown Figure 17, precipitation is observed in the samples of the preparation with ethanol, propanol and polymethylsilaxane liquid as solvents, indicating the instability of the components and thus the unsuitability of these solvents for the preparation of a nanoscale composite protective coating for car paint and varnish coatings. This is also confirmed by dynamic light scattering (Figure 19).

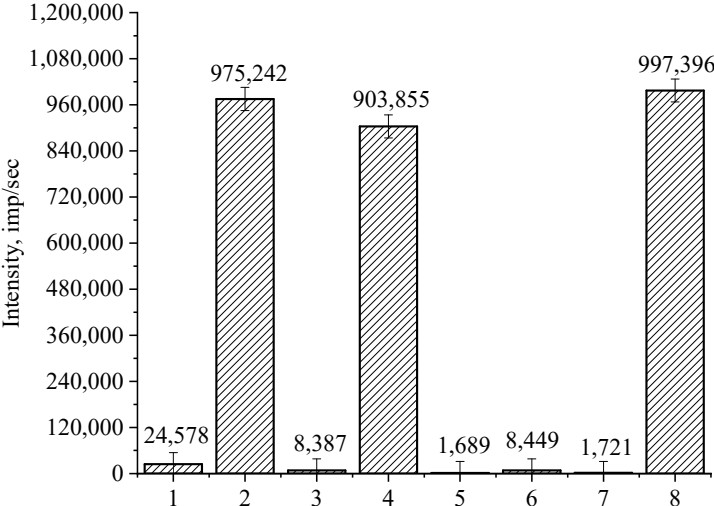

**Figure 19.** The intensity of light scattering of samples of the preparation for the formation of a nanoscale composite protective coating obtained using various solvents: (1) isopropanol, (2) ethanol, (3) n-butanol, (4) polymethylsiloxane liquid, (5) isoamyl alcohol, (6) benzyl alcohol, (7) amyl alcohol, (8) propanol.

It was found that in the samples of the preparation with ethanol, propanol and polymethylsilaxane liquid as solvents, the intensity of the light scattering takes values of 900,000 to 1,000,000 pulses/s, indicating the formation of a dispersed system with large particle aggregates. For the remaining samples, the light scattering intensity does not exceed 25,000 pulses/s, indicating that a dispersed system with large particle aggregates is not formed. As a result of the analysis of experimental data, we found that isopropanol is the optimal solvent, which is characterized by low cost, high safety, and ensures the solubility of all components without changing their physicochemical properties.

### 3.3. Optimization of the Preparation Technique for the Formation of a Nanoscale Composite Protective Coating

A multifactorial experiment was conducted to optimize the preparation method for the formation of a nanoscale composite protective coating. The following parameters were considered as input parameters: the volume concentration of plant resin, the volume concentration of tetraethoxysilane and the volume concentration of titanium tetraisopropylate. The wetting contact angle (θ) was considered as the output parameter. As a result of statistical processing of the data obtained, three-dimensional surfaces of the response of the output parameter θ from the input parameters are formed and presented in Figures 20–22.

We found that at concentrations of plant resin < 5 vol.% and tetraethoxysilane < 10 vol.% No hydrophobic surface is formed and the wetting contact angle of the formed surfaces is less than 60°. At concentrations of plant resin > 9 vol.% and tetraethoxysilane concentrations > 50 vol.%, the hydrolysis process of the latter is probably disrupted and the formation of a surface with a wetting contact angle more than 120° also does not occur. As can be seen from Figure 20, there are 2 extremums on the response surface at concentrations of plant resin < 1 vol.% and tetraethoxysilane > 40 vol.%, and at concentrations of tetraethoxysilane < 10 vol.% and plant resin > 8 vol.%, that allow formation of a coating with a contact wetting angle more than 120°.

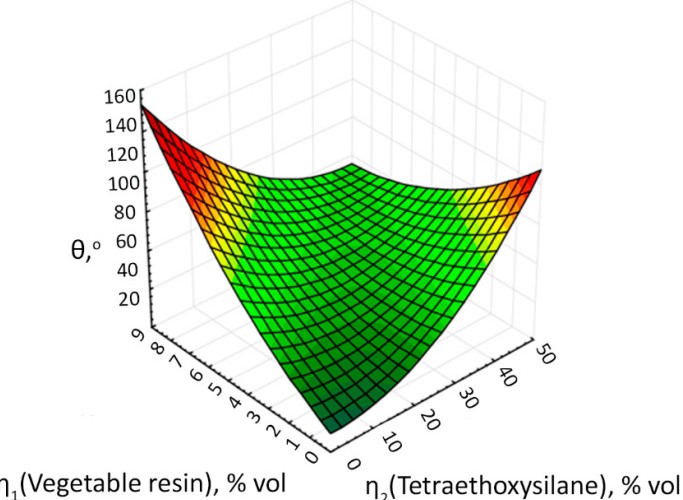

**Figure 20.** The response surface of the output parameter θ depending on the volume concentrations of plant resin and tetraethoxysilane.

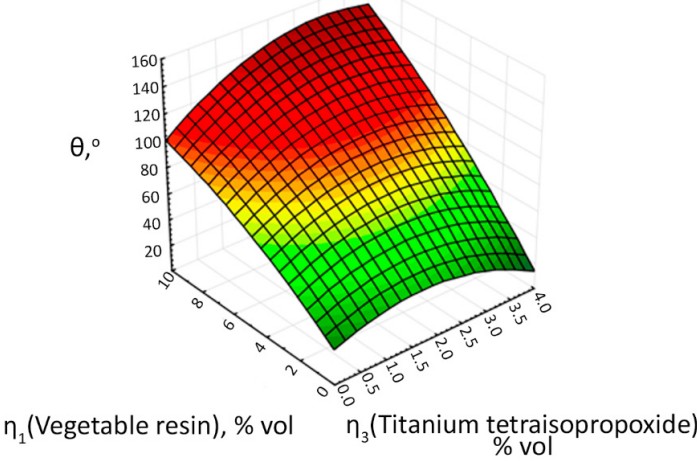

**Figure 21.** The response surface of the output parameter θ depending on the volume concentrations of plant resin and titanium tetraisopropylate.

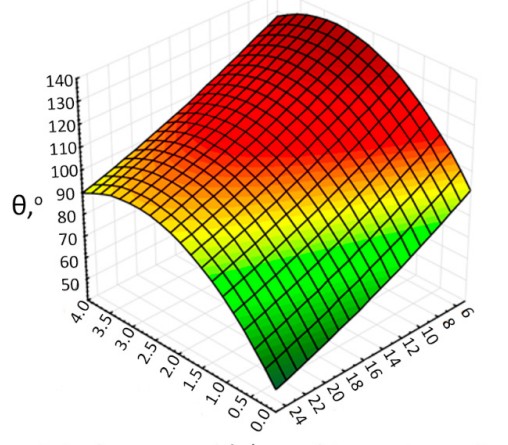

**Figure 22.** The response surface of the output parameter θ depending on the volume concentrations of tetraethoxysilane and titanium tetraisopropylate.

The analysis of the second response surface (Figure 21) shows the opposite situation, where the wetting contact angle increases and reaches a value of more than 120° at concentrations of plant resin > 8 vol.%, and at concentrations of titanium tetraisopropylate > 1 vol.%.

When analyzing the last response surface shown in Figure 22, it was found that the required wetting contact angle of more than 120° occurs at concentrations of tetraethoxysilane $\leq$ 18 vol.% and concentrations of titanium tetraisopropylate $\geq$ 2 vol.%

The analysis of the obtained data showed that the component composition of the preparation for the formation of a nanoscale composite protective coating significantly affects its hydrophobic properties, where it is possible to form a surface with a certain roughness and morphology [46]. The optimal component composition of the preparation for the formation of a nanoscale composite protective coating with hydrophobic properties and a wetting contact angle greater than 120° is: tetraethoxysilane $\leq$ 10 vol.%, titanium tetraisopropylate $\geq$ 2 vol.% and plant resin $\geq$ 8 vol.%

*3.4. Investigation of the Stability of a Nanoscale Composite Protective Preparation*

Following the study of the stability of nanoscale composite protective preparation for car paint and varnish coatings, the subsequent results were obtained, presented in Table 5.

**Table 5.** Results of the study of the stability of nanoscale composite protective preparation for car paint and varnish coatings.

| # | Humidity, % | Temperature, °C | Wetting Contact Angle before Experiment, ° | Wetting Contact Angle after Experiment, ° |
|---|---|---|---|---|
| 1 | 10 | −30 | 120 ± 1 | 120 ± 1 |
| 2 | 10 | 20 | 119 ± 1 | 119 ± 1 |
| 3 | 10 | 70 | 120 ± 1 | 120 ± 1 |
| 4 | 50 | −30 | 120 ± 1 | 120 ± 1 |
| 5 | 50 | 20 | 119 ± 1 | 119 ± 1 |
| 6 | 50 | 70 | 121 ± 1 | 121 ± 1 |
| 7 | 90 | −30 | 120 ± 1 | 120 ± 1 |
| 8 | 90 | 20 | 119 ± 1 | 118 ± 1 |
| 9 | 90 | 70 | 120 ± 1 | 120 ± 1 |

From the analysis of the obtained data, we found that no statistically significant changes in the wetting contact angle was detected in a humidity range of 10%–90% and a temperature range of −30–70 °C. It means that the developed nanoscale composite protective preparation will have no climatic restrictions during its operation and can be applied in different countries of the world with different weather conditions.

*3.5. Practical Approval of Nanoscale Composite Protective Preparation for Car Paint and Varnish Coatings*

Pilot production of the preparation for formation of nanoscale composite protective coating was carried out on the basis of LLC "Research and Production Company PRIDE" (Stavropol, Russia) according to Section 2.2.6. Figures 23–28 show photos of application of the preparation synthesized according to Section 2.6.

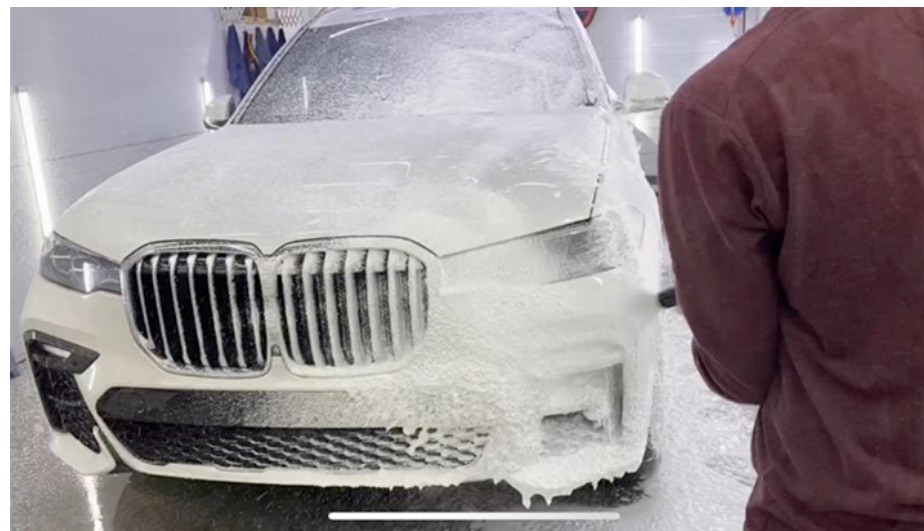

**Figure 23.** Two-phase surface washing.

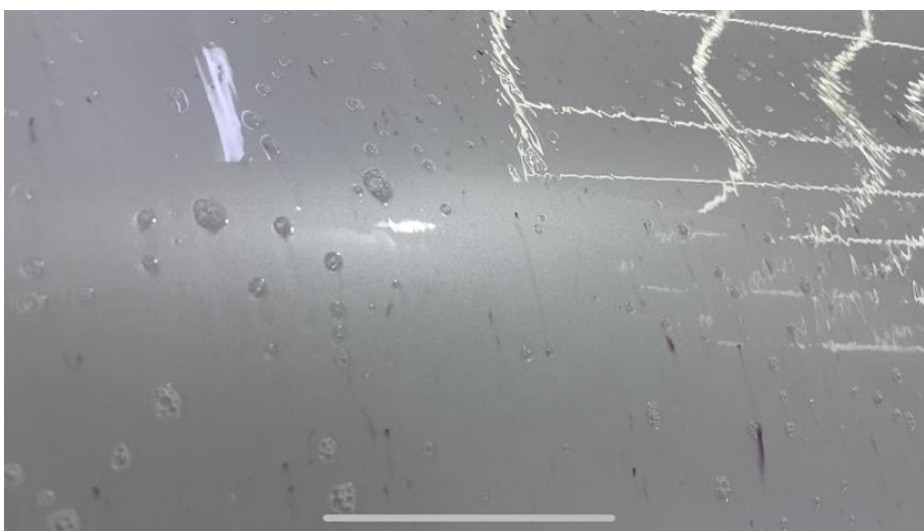

**Figure 24.** Removal of metal particles.

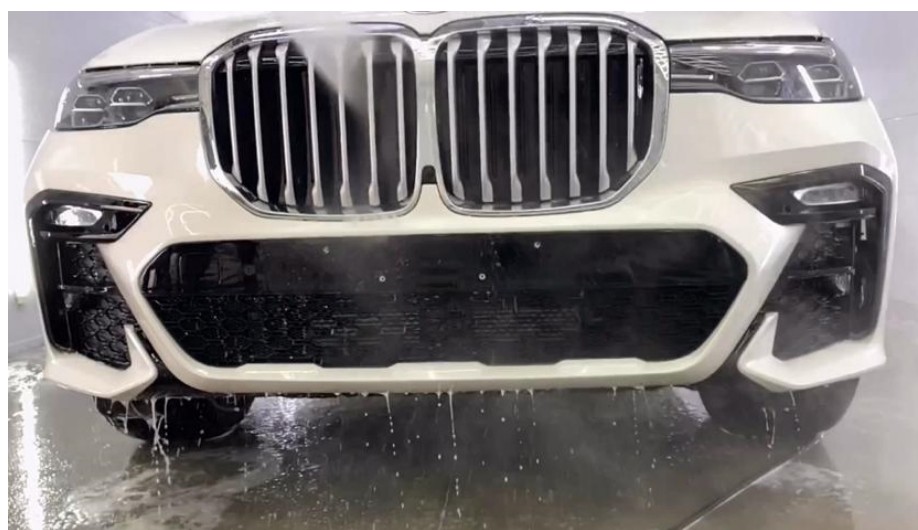

**Figure 25.** Car washing.

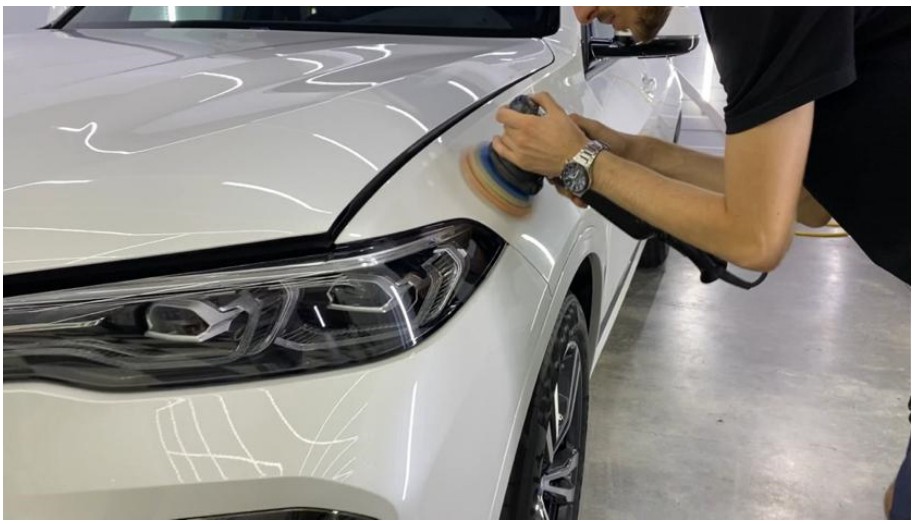

**Figure 26.** Polishing the car paintwork.

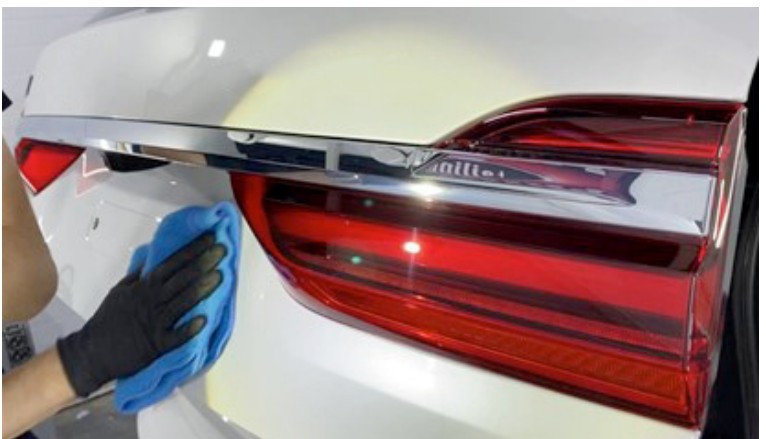

**Figure 27.** Degreasing the car paintwork.

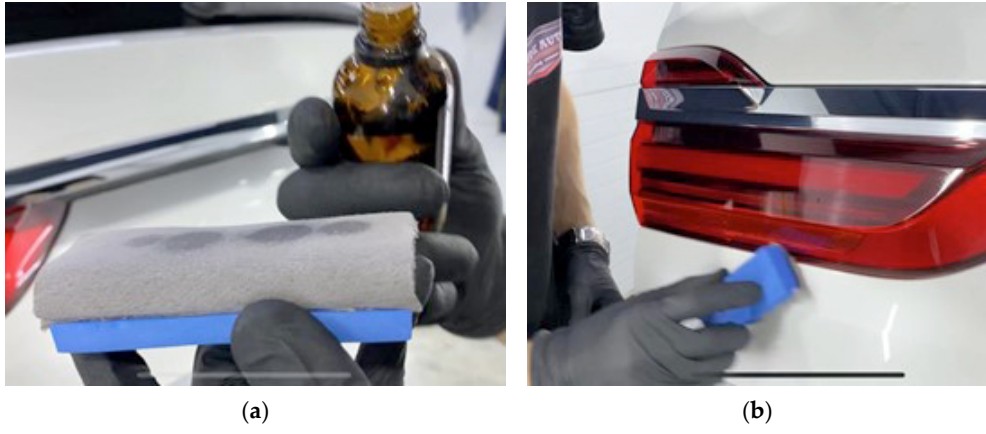

| (**a**) | (**b**) |

**Figure 28.** Application of the preparation for the formation of a nanoscale composite protective coating: (**a**) applying the preparation to the sponge, (**b**) applying the preparation to the car paint and varnish coating.

The contact angle of wetting an osmotic water droplet on the surface of the car paint and varnish coating with a nanoscale composite protective preparation was determined. One of the obtained photos is shown in Figure 29.

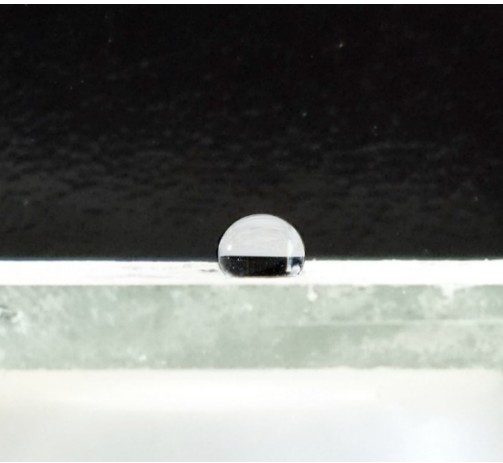

**Figure 29.** Photo of an osmotic water drop on the surface of the car paint and varnish coating with a nanoscale composite protective preparation.

Analysis of the obtained photos showed that the wetting contact angle on the paint and varnish of the treated car was greater than 120°, indicating that the developed nanoscale protective composite preparation has hydrophobic properties.

In the final stage of the experiment, the stability of the nanoscale protective composite preparation to the washing process was evaluated. The dependence of the wetting contact angle on the number of washing cycles is obtained and shown in Figure 30.

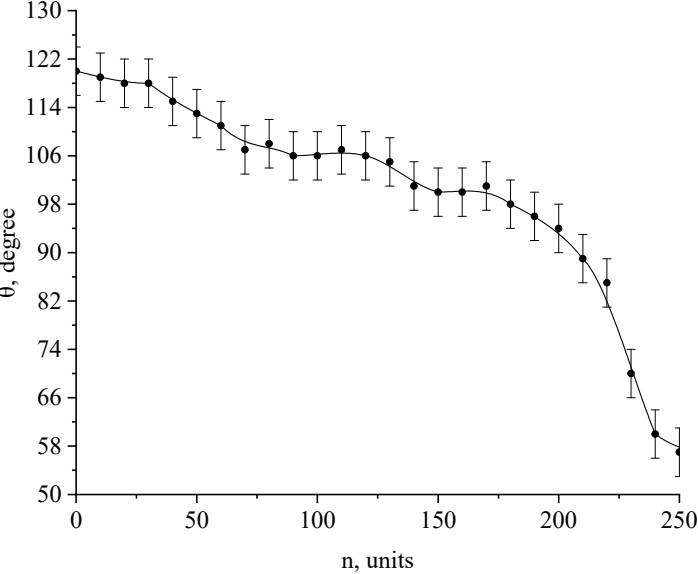

**Figure 30.** Dependence of the contact angle of wetting on the number of washing cycles.

Following data analysis, it was found that the developed nanoscale composite protective coating is able to maintain its hydrophobic properties ($\theta \geq 90°$) for more than 150 wash cycles. This indicates that the developed preparation has high potential for industrial use as protective composite coatings [43].

## 4. Conclusions

We developed the synthesis methods of $SiO_2$-$TiO_2$, $TiO_2$-$ZrO_2$, $SiO_2$-$ZrO_2$ and $SiO_2$-$TiO_2$-$ZrO_2$ nanocomposites. Phase composition analysis showed that the $SiO_2$-$TiO_2$, $SiO_2$-$ZrO_2$ and $SiO_2$-$TiO_2$-$ZrO_2$ nanocomposites samples have an amorphous structure, and the $TiO_2$-$ZrO_2$ samples are crystalline and contain non-stoichiometric zirconium oxide

with a tetragonal crystal lattice, as well as tetragonal modified titanium dioxide in an anatase-like structure.

A study of the stability of nanocomposites samples and their components in alkaline media showed that $SiO_2$-$TiO_2$, $TiO_2$-$ZrO_2$, and $SiO_2$-$ZrO_2$ nanocomposites samples in alkaline media are subject to dissolution, and $SiO_2$-$TiO_2$-$ZrO_2$ nanocomposite samples do not change, which is associated with the formation of solid solutions alongside unique physicochemical properties.

Based on the research conducted, we determined that the optimal nanocomposite to achieve a nanoscale composite protective coating is a $SiO_2$-$TiO_2$-$ZrO_2$ nanocomposite with a titanium dioxide content of 8% to 9.5% and zirconium dioxide from 0.5% to 2%, which can form a uniform structure on the surface of the paint and varnish coating and protect the car surface from exposure to ultraviolet radiation. Isopropanol is determined as the optimal solvent, which is characterized by low cost, high safety and ensures the solubility of all components without changing their physico-chemical properties.

The method of synthesis of the preparation for the formation of a composite protective coating at the nanoscale was optimized. We found that the component composition of the preparation for the formation of a nanoscale composite protective coating significantly affects its hydrophobic properties, varying which it is possible to form a surface with a certain roughness and morphology. The optimal component composition of the preparation for the formation of a nanoscale composite protective coating with hydrophobic properties and a wetting contact angle greater than 120° is represented by tetraethoxysilane $\leq$ 10 vol.%, titanium tetraisopropylate $\geq$ 2 vol.% and plant resin $\geq$ 8 vol.%

The main characteristic of the obtained coating is the composition, comprising titanium dioxide, silicon and zirconium, allowing to reach the most favorable structure, which facilitates to attain a contact wetting angle of the drop with a treated surface of 120°, but in this way, it is possible to achieve important values.

To test the developed preparation, we conducted an experiment with a BMW X6. The wetting contact angle on the treated paint of BMW X6 was greater than 120°, indicating the hydrophobic properties of the developed nanoscale composite protective preparation. Evaluation of the stability of the nanoscale composite protective coating to the washing process showed that the developed preparation is able to maintain its hydrophobic properties for more than 150 washing cycles. This indicates that the developed preparation has high potential for industrial use as protective composite coatings.

**Supplementary Materials:** The following supporting information can be downloaded at: https://www.mdpi.com/article/10.3390/coatings12091267/s1, Figure S1. Diffractograms of $TiO_2$-$ZrO_2$ nanocomposite samples: 1—0.1% $ZrO_2$, 2—0.5% $ZrO_2$, 3—1% $ZrO_2$, 4—2% $ZrO_2$, 5—3% $ZrO_2$; Figure S2. Diffractograms of $SiO_2$-$TiO_2$ nanocomposite samples: 1—10% $SiO_2$ and 90% $TiO_2$; 2—30% $SiO_2$ and 70% $TiO_2$; 3—50% $SiO_2$ and 50% $TiO_2$; 4—70% $SiO_2$ and 30% $TiO_2$; 5—90% $SiO_2$ and 10% $TiO_2$; Figure S3. Diffractograms of $SiO_2$-$TiO_2$-$ZrO_2$ nanocomposite samples: 1—0.1% $ZrO_2$ and 9.9% $TiO_2$; 2—0.5% $ZrO_2$ and 9.5% $TiO_2$; 3—1% $ZrO_2$ and 9% $TiO_2$; 4—2% $ZrO_2$ and 8% $TiO_2$; 5—3% $ZrO_2$ and 7% $TiO_2$; Figure S4. Diffractograms of $SiO_2$-$ZrO_2$ nanocomposite samples: 1—0.1% $ZrO_2$, 2—0.5% $ZrO_2$, 3—1% $ZrO_2$, 4—2% $ZrO_2$, 5—3% $ZrO_2$.

**Author Contributions:** A.V.B.: conceptualization, methodology, formal analysis, supervision; A.A.N.: formal analysis, writing—review and editing, resources and project administration; L.P.A.: methodology, investigation; V.N.V.: investigation and visualization; O.V.K.: formal analysis and writing—original draft; A.A.G.: validation, software and investigation; A.B.G.: investigation and visualization; A.A.B.: formal analysis and writing—original draft; D.G.M.: investigation and software; D.D.F.: investigation; V.A.L.: sample preparation and investigation; E.D.N.: sample preparation and investigation; M.A.S.: writing—review and editing. All authors have read and agreed to the published version of the manuscript.

**Funding:** This research received no external funding.

**Institutional Review Board Statement:** Not applicable.

**Informed Consent Statement:** Not applicable.

**Data Availability Statement:** All data are available upon request to the corresponding author.

**Acknowledgments:** The work was carried out using the equipment of the Center for Collective Use of the North Caucasus Federal University with financial support from the Ministry of Education and Science of Russia, unique project identifier RF—2296.61321X0029 (agreement No. 075-15-2021-687).

**Conflicts of Interest:** The authors declare that they have no known competing financial interests or personal relationships that could have appeared to influence the work reported in this paper.

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
