# Peer review of "Nanoscale Composite Protective Preparation for Cars Paint and Varnish Coatings"

_coatings, doi:10.3390/coatings12091267_

Round 1

Reviewer 1 Report

-          At Introduction, authors must write about the sol-gel method. Please, explain why this method was used in these experiments.

-          At 2.2.1., page 2, modify “dissolution of TEOS” with “dissolution of tetraethoxysilane”.

-          Schematic representation of the obtained materials must be added.

-          At page 12, modify “sample with 10% TiO2” with “sample with 10% TiO2

-          At Conclusions, authors mention: ”tetraethoxysilane ≤ 10% vol, titanium tetraisopropylate ≥ 2% vol and vegetable gum ≥ 8% vol.”. At section 2, it is mention “plant gum”. Please, specify which name is correct: vegetable or plant?

Author Response

We are grateful to the Reviewer 1 for his/her positive evaluation and for the time devoted to review our manuscript. All comments were useful and pleased us with the high level of understanding of the topic. We have addressed all recommendations as requested. All changes in the manuscript are marked by green. Red text is a result of English quality correction. Please see the point-by-point answers below

At Introduction, authors must write about the sol-gel method. Please, explain why this method was used in these experiments.

Thank you for recommendation, we added information about the sol-gel method to introduction

At 2.2.1., page 2, modify “dissolution of TEOS” with “dissolution of tetraethoxysilane”. Corrected, thank you!

-  Schematic representation of the obtained materials must be added.

Thank you very much for recommendation. We created the scheme of the preparation synthesis. We agree that now readers will easier understand how we prepared our samples

At page 12, modify “sample with 10% TiO2” with “sample with 10% TiO2

  Corrected, thank you!

At Conclusions, authors mention: ”tetraethoxysilane ≤ 10% vol, titanium tetraisopropylate ≥ 2% vol and vegetable gum ≥ 8% vol.”. At section 2, it is mention “plant gum”. Please, specify which name is correct: vegetable or plant?

We checked the whole text to use only one definition – plant resin

Reviewer 2 Report

The study of this manuscript is very interesting for readers and researchers. Overall presentation of work in a proper sequence. However, some clarification and improvements are highly needful in consideration of scientific output and explanations. My suggestions are mentioned below:

1. Authors used plant gum for the synthesis and development of car paint for coatings. Provide the details about plant from which gum was taken and specify the chemical compositions including physicochemical properties. It will be attractive to readers and reaserchers. 

2. The quality of figures is very poor. All figures should be revised with high resolution with containing scientific informations.

3. It is not clear to me how the mix-metallic nanoparticles were disperse in the plant gum? What was dispersity ratio? 

4. It will be better if authors will provide details of coating solution compositions using XPS and EDS analysis.

5. Authors should evaluate coating stability natural humidity, temperatures and acidic pHs conditions  including adverse combination of these factors.

6. What are key features in coating solution to provide the hydrophobicity? 

7. English should be improved in revised version of manuscript. 

Author Response

We are grateful to the Reviewer 2 for his/her positive evaluation and for the time devoted to review our manuscript. All comments were useful and pleased us with the high level of understanding of the topic. We have addressed all recommendations as requested. All changes in the manuscript are marked by green. Red text is a result of English quality correction. Please see the point-by-point answers below

Authors used plant gum for the synthesis and development of car paint for coatings. Provide the details about plant from which gum was taken and specify the chemical compositions including physicochemical properties. It will be attractive to readers and reaserchers.

Thank you very much for recommendation. We added technical information about plant resin in Table 1

The quality of figures is very poor. All figures should be revised with high resolution with containing scientific informations.

Thank you for useful comment. We have done our best to improve the quality of the figures. 

It is not clear to me how the mix-metallic nanoparticles were disperse in the plant gum? What was dispersity ratio? 

Thank you for your comment, we modified section 2.2.5, now it should be fro explanative. Dispersion was carried out using Ultrasonic homogenizer UP400S (Hielscher Ultrasonics GmbH, Teltow, Germany) with set parameters: Sono-rod H3 (titanium, 100 mm), pro-cessing power 200 W, frequency 24 kHz, amplitude 60%, pulsation 70%, and treatment duration 30 – 60 seconds. The achieved level of particle dispersion was 50-150 nm

 It will be better if authors will provide details of coating solution compositions using XPS and EDS analysis.

Thank you very much  for recommendation. Results of the EDS of our nanocomposites were presented and discussed in our previous work. We added information about carried out EDS analysis and gave a references, which we hope will satisfy readers. Also we added a short comment about the general result of EDS,

Authors should evaluate coating stability natural humidity, temperatures and acidic pHs conditions  including adverse combination of these factors.

Thank you for recommendation. We carried out additional experiment and added relevant results and data to Materials and Methods and Results and Discussion sections

What are key features in coating solution to provide the hydrophobicity? 

Thank you for your question. We added corresponding information to Conclusion

English should be improved in revised version of manuscript.

Done (correction are marked by red)

Round 2

Reviewer 1 Report

Dear Sirs,

The manuscript was improved and it can be published in this form.

Reviewer 2 Report

Authors improved manuscript significantly according to the suggested points. Thus, present form of manuscript can be accepted for publications.